# A Data Normalization Technique for Detecting Cyber Attacks on UAVs

Elena Basan [1,*] , Alexandr Basan [1] , Alexey Nekrasov [1] , Colin Fidge [2] , Evgeny Abramov [1] and Anatoly Basyuk [1]

1 Institute for Computer Technologies and Information Security, Southern Federal University, Chekhova 2, 347922 Taganrog, Russia
2 Faculty of Science, Queensland University of Technology (QUT), Gardens Point Campus, Brisbane, QLD 4001, Australia
* Correspondence: ele-barannik@yandex.ru; Tel.: +7-951-20-5488

**Abstract:** The data analysis subsystem of an Unmanned Aerial Vehicle (UAV) includes two main modules: a data acquisition module for data processing and a normalization module. One of the main features of an adaptive UAV protection system is the analysis of its cyber-physical parameters. An attack on a general-purpose computer system mainly affects the integrity, confidentiality and availability of important information. By contrast, an attack on a Cyber-Physical System (CPS), such as a UAV, affects the functionality of the system and may disrupt its operation, ultimately preventing it from fulfilling its tasks correctly. Cyber-physical parameters are the internal parameters of a system node, including the states of its computing resources, data storage, actuators and sensor system. Here, we develop a data normalization technique that additionally allows us to identify the signs of a cyber-attack. In addition, we define sets of parameters that can highlight an attack and define a new database format to support intrusion detection for UAVs. To achieve these goals, we performed an experimental study of the impact of attacks on UAV parameters and developed a software module for collecting data from UAVs, as well as a technique for normalizing and presenting data for detecting attacks on UAVs. Data analysis and the evaluation of the quality of a parameter (whether the parameter changes normally, or abrupt anomalous changes are observed) are facilitated by converting different types of data to the same format. The resulting formalized CPS model allows us to identify the nature of an attack and its potential impact on UAV subsystems. In the future, such a model could be the basis of a CPS digital twin in terms of security. The presented normalization technique supports processing raw data, as well as classifying data sets for their use in machine learning (ML) analyses in the future. The data normalization technique can also help to immediately determine the presence and signs of an attack, which allows classifying raw data automatically by dividing it into different categories. Such a technique could form the basis of an intrusion detection system for CPSs. Thus, the obtained results can be used to classify attacks, including attack detection systems based on machine learning methods, and the data normalization technique can be used as an independent method for detecting attacks.

**Keywords:** UAV; CPS; GPS; cyber threats; anomalies; spoofing; entropy; cyber-attacks; data collection; parameters; ontology

## 1. Introduction

A Cyber-Physical System (CPS) provides a close connection between the cyber and physical domains by embedding cyber processes (e.g., communication, computing or control) into physical devices. Today, safety-critical CPSs and Unmanned Aerial Vehicles (UAVs) are widely used, so ensuring their secure operation is an important task.

UAV Intrusion Detection Systems (UAV-IDSs) are being developed to detect anomalous behavior or unexpected activities in networks by automatically analyzing their behavior based on a given hypothesis and/or policies that are managed by the given network's

security rules [1]. A UAV-IDS monitors the system's configuration, data files and/or network transmissions to check if an attack is present. Therefore, a UAV-IDS is an essential first step in preventing any covert/overt actions aimed at exploiting UAV security vulnerabilities in order to cause the system to fail or to hijack its operation. Such misuse can be defined as any undesirable action that may cause any harm in terms of the performance or safety of an entire UAV group. Attacks exploit vulnerabilities in UAV systems, which can result from the misconfiguration of UAV networks, implementation errors and incorrect design and/or protocols [2].

Successful UAV intrusion detection relies on a data analysis subsystem whose main tasks are as follows:

- Collection of data from UAV subsystems and sensors;
- Data aggregation for further analysis;
- Normalization of collected data;
- Recording data in a format convenient for further use.

The advantages of a UAV data analysis subsystem is the use of a single data set for solving various protection tasks, as well as a reduction in the number of software module calls needed to access data from hardware devices. Firstly, it provides greater reliability, given that a UAV is a complex system controlled by multiple processes, making it necessary to consider different processes within a single control system [3]. When building a protection system, each process must be authorized, and its access to the hardware must be controlled and fixed to avoid system malfunctions. Therefore, it is easier to manage these events from a security point of view when there are fewer such calls. Secondly, the tasks of program modules using already prepared data sets speed up information processing and decision making, which increases response times and the overall performance of the system [4].

The challenge is that a potential adversary can carry out targeted attacks on certain cyber-physical parameters or can carry out attacks that indirectly affect the physical properties of the system. For example, an attack may try to deplete the device's battery or overflow the communications network with false requests to connect to a node, which may be a UAV [5]. Moreover, the effects of different attacks on system parameters may be hard to distinguish. For example, multiple kinds of attack may cause the battery to drain faster. On the other hand, some kinds of attack have distinctive signatures. For example, an overflow attack using false requests affects the volume of network traffic [6]. In general, attacks can be classified as active and passive, internal and external, etc. In this study, we distinguish the following types of attacks:

- Attacks on integrity: This kind of attack leads to changes in the original information or the initial state of the system, which can produce distortions of information and violations of the properties of the system [7].
- Attacks on availability: These attacks block the operation of services, resources and executive mechanisms, or they prevent the ability to access necessary resources [8].
- Attacks on confidentiality: These attacks are aimed at obtaining information about the system. They do not implement any active actions aimed at violating the functionality of the system, but they only receive valuable information about it [9].
- Resource exhaustion attacks: Such attacks aim to increase the use of UAV resources. Although UAV functionality may not be violated, power consumption increases [10].
- Access attacks: These are similar to confidentiality attacks, but instead of passively obtaining information about the system, an access attack actively intrudes the system to obtain intelligence information [11].

Many previous researchers have used UAV system data as a basis for intrusion detection. To support the fight against the malicious misuse of networks, an algorithm called AMDES (Unmanned Aerial Detection System with Multifractal Analysis) was proposed by Zhang et al. to detect spoofing attacks [12]. Their algorithm is based on both wavelet leader multifractal analysis (WLM) control and machine learning (ML) control. Such estimation is

used to detect and describe abnormal flooding that can be observed in a network of UAVs. A simulation environment was built using radar system records. A total of 31 daily radar records were processed, covering one month of activity. Each original recording contained about 800,000 samples. The entire original recording was used to create 36 versions of the trace with a certain penetration level. To exclude a significant overbalance of "normal" data sets versus the attacked set, several traces containing a miniature attack (1 A/C, 1–10%) were also included in the training set.

Aissou et al. demonstrated the collection of data when attacks do and do not occur [13,14]. The hardware that was used was a Universal Software Radio Peripheral (USRP). Open source GNSS-SDR (Global Navigation Satellite System–Software-Defined Radio) software based on GNU was used. Satellites were emulated by setting the following parameters: signal-to-noise ratio, Doppler shift, number of satellites, etc. To emulate a GPS (Global Positioning System) spoofing attack, three cases were considered. In the first case, the attacker does not know the exact coordinates of the drone and randomly generates a signal. In the second case, the attacker knows the position of the UAV and deliberately carries out the attack. In the third case, several synchronously tuned antennas were in use. Attacks were carried out by means of emulation and, accordingly, had no effect on the analyzed parameters. Aissou et al. used two options to normalize the data collected from the experiment. The first option included the calculation of the Spearman's Correlation Coefficient, and Non-Stationary Data Modification [13]. In the second option, they changed the normalization method and used the Spearman's correlation method after applying the Min–Max technique. Accordingly, they used various machine learning methods, choosing the best one for attack detection. This technology for collecting and normalizing data was also used by Khoei et al. [15]. The advantage of their work is that they considered several options for GPS spoofing attacks, which increased the level of attack detection. On the other hand, to implement the IDS, the UAV must carry out an analysis based on neural networks, which requires significant computing power, and this approach detects GPS spoofing attacks only.

Whelan et al. proposed analyzing UAV flight logs and highlighting the signs of an attack from them [16]. Moreover, in contrast to previous works, they proposed dividing the features into separate categories and using a separate neural network for analysis for each category. Feature separation, according to the authors, not only makes it possible to detect attacks but also to potentially classify the type of attack or the target sensor. They divided the flight log into several CSVs depending on the sensor/topic on the UAV. However, topics may be polled at different rates, and so certain values must also be interpolated. Some features were grouped only for the GPS navigation subsystem. These features included latitude, longitude and altitude, as well as speed and location data. Based on this concept, the authors suggested that different attacks affect different feature clusters. In fact, when an attack is carried out, it often affects several types of data at once. For example, if the UAV makes an emergency landing during a jamming attack, as a DJI Mavic Air drone does, its coordinates are also changed, so the intrusion detection system may falsely determine that two attacks are being made. Another problem with this study is how the authors presented the attack pattern. They assumed that GPS sensor data are received by the autopilot at a frequency of 5 Hz. However, when fake signals are injected into the simulation environment, they are sent at a much higher rate. This causes the autopilot to switch to a stronger signal [17]. After that, the assessment of the position of the UAV changes, which leads to a course deviation [18]. When the attack stops, the UAV locks onto the legitimate signal again, reverts to its correct position and resumes its original mission trajectory. In real conditions, however, when an attack is being carried out, such recovery does not proceed so smoothly. In a real environment, when a UAV flies in an open sky and receives a signal from real satellites, an attacker who has even a powerful antenna cannot accurately simulate the satellite signal, as there is additional noise. In addition, the number of satellites on which the UAV is fixed can change, so not only the coordinates but also the flight altitude can change [19]. Thus, the number of variable factors is much larger than

what Whelan et al. suggest [16]. After a detailed analysis of the data sets, for example, it has been shown that the number of satellites fixed during normal flight is somehow less than during a noise attack but more than during a spoofing attack [19]. In this case, there is no explicit transition from the normal state to the state of the attacked UAV, although it is often important to fix the emerging peak value of the parameter. Such a value can be considered a selection, and it is not considered when creating the data set.

Park et al. used the same data set while carrying out some transformations [20]. For example, they divided the signs of an attack into five even more enlarged categories. At the same time, they removed unnecessary features according to two rules. According to the first rule, all "non-universal" features were removed, and only those that are inherent in all UAVs were kept. The second rule implies the exclusion of parameters that have constant, unchanging values and parameters that can have missing values. If the first rule really allows unifying the data set, there are still questions about the second rule. For example, there are sensors that usually show a constant true/false value throughout a flight, but on the rare occasions when they do change, an emergency response is needed, so excluding them entirely is inappropriate. Nonetheless, the key advantage of the work is that methods for normalizing raw data are used. In particular, timestamp merging is applied. The authors unified the length of each feature using a pool of timestamps. They also used the Min–Max technique discussed earlier to bring the whole data set to a single scale.

Wang described an attack detection system based on neural networks using signature analysis [21]. To obtain a traffic signature in three dimensions (3D), he measured the scaling function (Zeta) against statistical moments ($q$), which can take positive or negative values, as well as against the traffic timescale. Then, using the wavelet multifractal analysis method based on the UAV hybrid network simulator, he obtained the necessary signatures. To obtain experimental data, he used a stand where normal TCP traffic is generated by five TCP sources that generate long TCP streams to the recipient through a router with different channel bandwidths. Two types of DDoS attacks were considered: Constant Flash-Crowd (CFC) and Progressive Flash-Crowd (PFC) attacks. These anomalies were generated using the HPing31 tool. In the scenario, HPing3 exchanges thousands of small TCP streams to generate a SYN flood attack on the receiving host. Considering what consequences may occur due to the attack in question, Wang obtained interesting results, especially in relation to the data set for neural network training and attack detection. Nevertheless, the question remains of how much this attack can affect the UAV and how easily it can be implemented. In fact, we can protect ourselves from SYN flood attacks even with standard operating system tools without resorting to specialized software. Although this type of attack is still relevant, not all modern devices are sensitive to it. In addition, it can only be implemented if the attacker is inside the network and carrying out preliminary reconnaissance actions. Detecting an attack by only one factor and attacks by only one traffic analysis factor may not be entirely productive.

Penglong et al. discussed another method for detecting spoofing attacks on UAVs [22]. They also used neural networks, testing several wireless network architectures and confirming the effectiveness of their method, with fairly fast detection and a low number of false positives. They used the following factors to detect attacks: Satellite Vehicle Number (PRN), Doppler Shift Measurement (DO), Pseudo Range (PD), Receiver Time (RX), Decoded Time Information (TOW), Carrier Phase Shift (CP), Prompt Correlator (PC), Late Correlator Output (LC), Early Correlator Output (EC), Prompt In-phase Prompt (PIP), Prompt Quadrature Prompt (PQP), Carrier Loop Doppler Measurements (TCD) and Signal-to-Noise Ratio (CN0). Again, when detecting an attack, the authors relied only on external factors and analyzed the environment around the UAV. They conducted the study in laboratory conditions, and it is not completely clear how a real environment would affect the detection of an attack. Nonetheless, as a result of the use of data normalization and preprocessing, this system is likely to be effective.

Thus, research in the field of detecting attacks on UAVs is quite active. Machine learning methods, especially neural networks, are used by most researchers aiming to

improve detection accuracy. In this case, they often use ready-made data sets collected and provided by others. However, not all studies focus on detecting the signs of an attack and on specific conditions and signals that may indicate that an attack is being carried out. In some cases, the data markup is based on laboratory tests and methodologies that collect data starting at exactly the time that the attack began, so this moment is considered the beginning of data collection during the attack. Such a simulation implies that the attack unambiguously affects certain system parameters and works immediately. Under real conditions, attacks can produce ambiguous effects on system parameters and can take time to be detectable, so a neural network trained on the same data is not able to detect an attack in real time. In addition, not all researchers use data normalization. The data must follow a uniform format for more efficient discovery. Some normalization methods even unhelpfully "clean" the data set of useful features, which could have contributed to attack detection if left intact, by removing outliers or interpolating the data.

Data sets for UAV attack detection are usually derived from two sources. In some research [12,21,22], the source of data is the external environment. The UAV must either listen to the radio frequency range or all traffic. Most likely, such attack detection takes place not on board the UAV, but by some third-party means. Perhaps, this is possible for small-sized UAVs that fly in a limited radius, but it is not clear that this can be applied to UAVs that fly hundreds of kilometers. In other studies [13–20], the authors use UAV flight logs as a data source. However, these raw data are not always normalized and reduced to a single consistent form. In addition, the simulation of an attack is usually based on a change in noise or coordinates, but the position of a UAV does not take into account how the attack affects its internal subsystems.

In our study, therefore, we developed a data normalization technique that simultaneously allows us to identify the signs of an attack. In addition, we defined sets of parameters that signal an attack and described a new database format for intrusion detection for UAVs. To achieve these goals, in this paper, we present:

- An experimental study of the impact of attacks on UAV parameters;
- A software module for collecting data from UAVs;
- A technique for normalizing and presenting data for detecting attacks on UAVs.

The contributions of this article are as follows. Firstly, an analysis of attacks that are most characteristic of UAVs is carried out, as a cyber-physical system. Cyber-physical parameters that are affected by various attack classes are determined. Secondly, the formalization and modeling of UAVs, as a CPS, through the analysis of the interconnection of UAV subsystems to cyber-physical parameters and their changes, is carried out. Thirdly, a technique for normalizing cyber-physical parameters is presented. Bringing various types of data to one format makes it easier to meet the conditions for the analysis of their changes and for evaluating the quality of the parameter (whether the parameter changes normally or sharp abnormal changes are observed). The formal description of the UAVs presented in the article as a CPS can become the basis for creating a digital twin of UAVs and researching it in terms of cyber security. The presented normalization technique can allow for processing raw data, as well as for classifying data sets for the use of ML methods in the future.

## 2. Materials and Methods

### 2.1. Ontological Model of the UAV Data Collection System for Attack Detection

In this study, we integrated an ontological approach to assess the impact of various attack categories on the cyber-physical parameters of a UAV. Then, the subsystems of the UAV were related to the cyber-physical parameters. This approach allows grouping the signs of an attack based on which subsystem they are intended for. To determine the parameters that are affected by an attack, we proceeded from real attacks that were carried out on a UAV and the consequences that arose in these cases. Furthermore, a systematization of the data obtained was carried out, as illustrated in Figure 1. The types of attacks are presented on the left. Attacks are grouped based on which security properties

they violate. Moreover, attacks can affect cyber-physical parameters both directly and indirectly, as shown in the diagram.

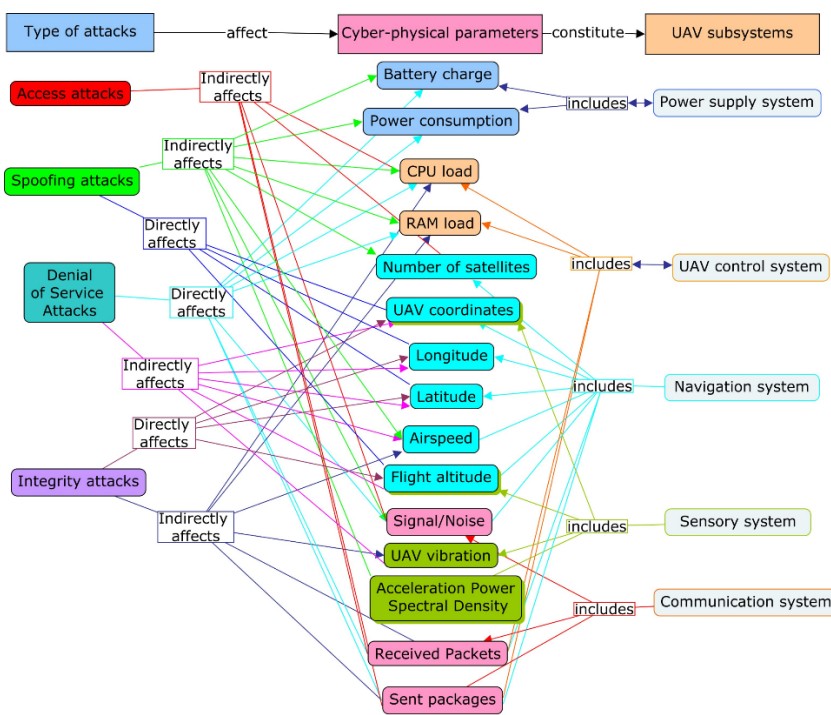

**Figure 1.** An ontological model for representing knowledge about the impact of attacks on the cyber-physical parameters of a UAV.

Direct influence is characterized by the fact that the final change in the parameters or behavior of the UAV is associated with the original purpose of the attack. For example, a GPS spoofing attack is aimed at changing UAV flight coordinates and moving it away from its target. Therefore, the direct goal of the attacker is to influence the following parameters: UAV coordinates, latitude, longitude and altitude (flight altitude is an optional parameter and may not be set by the attacker). GPS spoofing falls into the category of spoofing attacks, which is highlighted by a bright green rectangle. The direct impact of the GPS spoofing attack is marked by blue lines. With direct impact, it is clear why certain parameters are chosen, but with indirect parameters, ambiguities may arise.

Consider again the example of a GPS spoofing attack. The indirect effect of the attack in Figure 1 for the spoofing attack is highlighted with a bright green line. A GPS spoofing attack can indirectly affect such parameters as battery charge, power consumption, CPU load, RAM load, the number of satellites fixed, airspeed, signal-to-noise ratio and Acceleration Power Spectral Density. Let us analyze why an attack can indirectly affect these categories of parameters. Firstly, battery charge, power consumption, CPU load and RAM load change when attacked, so the UAV drastically changes its behavior. Even if we are not talking about a GPS spoofing attack, but about another kind of attack, it is not difficult to assume that the attacker's goal is to change the UAV's flight plan, and regardless of what data are faked, it is somehow related to an alteration in the UAV's mission. Moreover, if a GPS spoofing attack is carried out and the attacker seeks to displace the UAV by giving it a fake waypoint, then the UAV strives to move to the fake point as quickly as possible. When spending additional power on moving and processing data, a change in the groups of parameters listed above is observed. In this case, the attacker must use a more powerful antenna than the UAV itself to interrupt the real signal. In this regard, therefore, one of the indirect signs of the attack is a change in the signal-to-noise ratio. In addition, the number of satellites fixed by UAVs under such attack conditions does not correspond to the normal situation [23], so this parameter is also subject to change.

In Figure 1, the first column lists attack types, and the second column shows cyber-physical parameters. These concepts are related to each other in a one-way relationship. The cyber-physical parameters are also grouped according to which UAV subsystems, shown in the third column, are associated with them. However, some parameters can be influenced by several subsystems. In particular, the UAV's flight coordinates and its altitude are affected by several subsystems. These cyber-physical parameters can be measured both using the global navigation system and the UAV's internal sensors, such as the compass, barometer, etc. The set of parameters is universal and can be obtained from almost any flight controller. By dividing the parameters into groups and by attacks that affect the parameters, as well as the further formation of the attack vector, these sets can be adjusted.

### 2.2. Formalization of the Attack Model

In general, three broad classes of attacks can be distinguished. In this case, spoofing attacks are included in the class of integrity attacks, which corresponds to the classical classification. Spoofing attacks have been shown separately in Figure 1 to highlight which parameters they affect. Reconnaissance attacks are included in the class of access attacks because they essentially perform a similar function. The general goal of these attacks is to find an entry point to the system by examining the UAV's architecture. In the case of UAVs, the entry point is usually the communication channel. Since the UAV may be outside a controlled area, and since an attacker can physically intercept it, the UAV's interfaces and hardware can also become an entry point into the system. Table 1 presents the relationship among attacks, consequences and subsystems that are susceptible to attack and that allow the attack to be detected by analyzing changes in cyber-physical parameters, i.e., a connection is explicitly established between the attack and its influence on the cyber-physical parameters of the UAV.

**Table 1.** Category and comparison of attacks with the consequences and subsystems of UAVs.

| Attack Type $A_n$ | Attacks $N_i$ | Consequences $C_{mn}$ | | Subsystem $S_{jn}$ | |
|---|---|---|---|---|---|
| **Access attacks (Reconnaissance attacks) $AA_{n1}$** | Password brute force $AA_{i1}$ | - | Gaining access to the UAV network $CC_{m1}$ | - | Communication system $S_{j1}$ |
| | | - | Traffic interception $CC_{m2}$ | - | Control system $S_{j2}$ |
| | | - | Conducting further attacks on UAV $CC_{m3}$ | | |
| | | - | Access to UAV control $CC_{n4}$ | | |
| | RF analysis $AA_{i2}$ | - | $CC_{m2}$ | - | $S_{j1}$ |
| | | - | $CC_{m3}$ | | |
| | | - | Obtaining information about data channels $CC_{m5}$ | | |
| | | - | Obtaining information about the UAV $CC_{m6}$ | | |
| | Analysis of communication channels $AA_{i3}$ | - | $CC_{m2}$ | - | $S_{j1}$ |
| | | - | $CC_{m3}$ | | |
| | | - | $CC_{m5}$ | | |
| | Detection via physical channels $AA_{i4}$ | - | $CC_{m3}$ | - | $S_{j1}$ |
| | | - | $CC_{m5}$ | | |
| | | - | $CC_{m6}$ | | |

**Table 1.** *Cont.*

| Attack Type $A_n$ | Attacks $N_i$ | Consequences $C_{mn}$ | Subsystem $S_{jn}$ |
|---|---|---|---|
| | UAV physical interception $AA_{i5}$ | - $\{CC_{m1}: CC_{m6}\}$ | - Power supply system $S_{j3}$<br>- Sensory system $S_{j4}$<br>- $\{S_{j1}: S_{j5}\}$ |
| | Access to interfaces $AA_{i6}$ | - $\{CC_{m1}: CC_{m6}\}$ | - $\{S_{j1}: S_{j5}\}$ |
| Integrity attacks $IA_{n2}$ | Transmitted data modification $IA_{i1}$ | - Flight task change $CC_{m7}$<br>- Coordinate change $CC_{m8}$<br>- Target change $CC_{m9}$<br>- Changing flight parameters $CC_{m10}$<br>- Mission violation $CC_{m13}$<br>- Flight violation $CC_{m14}$<br>- Flight deviation $CC_{m15}$ | - Navigation system $S_{j6}$<br>- $S_{j1}$<br>- $S_{j2}$ |
| | False data injection $IA_{i2}$ | - Receiving false control commands $CC_{m11}$<br>- Receiving false telemetry $CC_{m12}$<br>- $CC_{m13}$<br>- $CC_{m14}$<br>- $CC_{m15}$ | - $S_{j1}$<br>- $S_{j2}$<br>- $S_{j6}$ |
| | Replay attack $IA_{i3}$ | - $CC_{m13}$<br>- $CC_{m14}$<br>- $CC_{m15}$ | - $S_{j1}$<br>- $S_{j2}$<br>- $S_{j6}$ |
| | Black hole attack $IA_{i4}$ | - $CC_{m7}$<br>- $CC_{m8}$<br>- $CC_{m9}$<br>- $CC_{m10}$<br>- $CC_{m13}$<br>- $CC_{m14}$<br>- $CC_{m15}$ | - $S_{j1}$<br>- $S_{j2}$<br>- $S_{j6}$ |
| | Message disclaimer $IA_{i5}$ | - $CC_{m11}$<br>- $CC_{m12}$ | - $S_{j1}$<br>- $S_{j2}$<br>- $S_{j6}$ |
| | Firmware modification $IA_{i6}$ | - UAV violation $CC_{m16}$<br>- $\{CC_{m1}: CC_{m16}\}$ | - $\{S_{j1}: S_{j5}\}$ |
| | Modification of hardware components $IA_{i7}$ | - $\{CC_{m1}: CC_{m16}\}$ | - $\{S_{j1}: S_{j5}\}$ |
| | Modification of operating system files $IA_{i8}$ | - $\{CC_{m1}: CC_{m16}\}$ | - $\{S_{j1}: S_{j5}\}$ |
| | UAV control source replacement $IA_{i9}$ | - $\{CC_{m1}: CC_{m16}\}$ | - $S_{j1}$<br>- $S_{j2}$<br>- $S_{j6}$ |

**Table 1.** *Cont.*

| Attack Type $A_n$ | Attacks $N_i$ | Consequences $C_{mn}$ | | Subsystem $S_{jn}$ | |
|---|---|---|---|---|---|
| **Denial of Service Attacks $DA_{n3}$** | Jamming the control channel $DA_{i1}$ | - | $CC_{m13}$ $CC_{m14}$ $CC_{m15}$ Data loss $C_{m17}$ Loss of control $C_{m19}$ UAV destruction $C_{m20}$ | - | $\{S_{j1}: S_{j5}\}$ |
| | Jamming the navigation channel $DA_{i2}$ | - | $CC_{m8}$ $CC_{m10}$ $CC_{m14}$ $CC_{m15}$ $C_{m20}$ | - | $\{S_{j1}: S_{j5}\}$ |
| | Request Flood Attack $DA_{i3}$ | - | $C_{m17}$ $CC_{m13}$ $CC_{m14}$ $CC_{m15}$ | - | $\{S_{j1}: S_{j5}\}$ |
| | Connection reset $DA_{i4}$ | - | UAV loss $C_{m18}$ $CC_{m13}$ $CC_{m14}$ $CC_{m15}$ $C_{m17}$ $C_{m19}$ | - | $\{S_{j1}: S_{j5}\}$ |
| | UAV physical accessibility violation $DA_{i5}$ | - | $\{CC_{m1}: CC_{m19}\}$ | - | $\{S_{j1}: S_{j5}\}$ |
| | Hijacking of aircraft $DA_{i6}$ | - | $\{CC_{m1}: CC_{m19}\}$ | - | $\{S_{j1}: S_{j5}\}$ |

In addition to the fact that Table 1 helps to identify clear signs of an attack, there is also a formalization of ideas about attacks on UAVs. As we can see from the table, different classes of attacks can affect the same subsystems and can lead to the same consequences.

Let us take a look at the attack model in detail. Attack $A_n$ includes one or more attacks from the set of attacks $N$ of one or more classes $\{AA_{ni}, IA_{ni}, DA_{ni}\}$, i.e., access, integrity and denial of service attacks, respectively. Each attack affects a subsystem $S_j$ or a set of subsystems $\{S_{j1}: S_{jm}\}$, which leads to one $C_m$ or several consequences from the set of consequences $\{CC_{m1}: CC_{m19}\}$.

Thus, an attack can be expressed in terms of a set of systems it affects and a set of consequences:

$$A_n = [\ \{S_{jn}\}; \{C_{mn}\}]　(1)$$

Using this approach, it is possible to determine by the totality of signs what kind of attack is being carried out. If we expand such a scheme and associate each subsystem with its related cyber-physical parameters, as shown in the ontological model, we can express the subsystem through a set of cyber-physical parameters that describe it:

$$S_{jn} = \{P_{i1}, \dots P_{in}\}　(2)$$

where $P_{in}$ is the cyber-physical parameter that belongs to the subsystem.

Thus, it is possible to express a subsystem through a set of parameters of different subsystems:

$$A_n = [\ \{P_{in}\}; \{C_{mn}\}]　(3)$$

At the same time, some cyber-physical parameters may be included in the same subsystems; for example, navigation and sensor systems may partially duplicate each other.

### 2.3. Raw Data Normalization Technique for Detecting Attacks on UAVs

Once an unambiguous relationship among attacks, consequences and cyber-physical parameters has been established, it is possible to assess the presence of an attack through cyber-physical parameters. Our methodology is based on the methods of probability theory and mathematical statistics. We define an attack in terms of the likelihood of consequences occurring by violating cyber-physical parameters, i.e., if we observe changes in cyber-physical parameters, there is a possibility that an attack is being carried out. As these changes become stronger, and depending on which and how many cyber-physical parameters are involved in the attack, the attack is more likely to be confirmed, and the probability of a particular attack being responsible can be determined. In this case, Bayes' theorem applies. Mathematically, Bayes' theorem shows the relationship between the probability of event $A$ and the probability of event $B$, $P(A)$ and $P(B)$, the conditional probability of the occurrence of event $A$ with existing $B$ and the occurrence of event $B$ with existing $A$, $P(A|B)$ and $P(B|A)$ [24]. For example, we need to determine the relationship between the probability of an attack, given a change in the parameters. Then, we can express the probability with the following equation:

$$P(A_n|P_n) = \frac{P(P_n|A_n)P(A_n)}{P(P_n)} \tag{4}$$

where $P(A_n)$ is the a priori probability of the occurrence of an event that is described as an attack, $P(A_n|P_n)$ is the probability of an attack $A$ occurring when parameter $P$ changes (a posteriori probability), $P(P_n|A_n)$ is the probability of changing parameter $P$ when attack $A$ occurs and $P(P_n)$ is the total probability of the occurrence of a change in parameter $P$.

Specifically, we believe a priori that an attack has occurred, and we need to understand which parameters are affected and with what probability they indicate its occurrence. In problems and statistical applications, $P(P_n)$ is usually calculated using the formula for the total probability of an event depending on several inconsistent hypotheses that have a total probability. In our case, as a rule, the attack depends on changing several parameters at once, so it is rational to use the following equation:

$$P(P_{in}|A_n) = \frac{P(A_n|P_{in})P(P_{in})}{\sum_{j=1}^{N} P(A_n|P_{jn})P(P_{in})P(P_{jn})} \tag{5}$$

From Bayes' theorem and probability, in the future, we can proceed to the construction of a Bayesian classifier to determine the types of attacks by changing parameters. The Bayesian classifier belongs to the ML category of algorithms. The bottom line is this: the system, which is faced with the task of determining whether the next change in the parameter is anomalous, has been trained in advance by some amount of data, in which it is already determined where the anomaly is, and where the normal behavior is. It has already become clear that this is training with a teacher, where we act as a teacher. The Bayesian classifier represents the data set that we receive from the UAV (in our case, these are changes in cyber-physical parameters) in the form of a set of metrics that supposedly do not depend on each other (this is where the naivety comes from). It is necessary to calculate the score for each class (normal behavior/attack spoofing attack silencing) and to choose the one that turned out to be the maximum.

Thus, it is necessary to record the fact of a change in cyber-physical parameters and, based on this, to determine the probability of the cause being an attack. For this, we use probability distributions. Table 2 presents the basic equations and the sequence of calculations necessary for data normalization. In this study, two types of distributions were used: $\chi^2$ and Poisson distributions. The $\chi^2$ test is used to determine whether a hypothesis is supported by an experiment.

**Table 2.** Methodology for normalizing raw data on changing cyber-physical parameters of UAVs.

| No. | Name of Equation | Equation | Description |
|---|---|---|---|
| 1 | Average value for a cyber-physical parameter in the range of a sliding window $\overline{P_{in}}(\Delta w)$ | $\overline{P_{in}} = \frac{1}{n} \sum\limits_{j=1}^{n} P_{ij}$ | $n$ is the sample size, $P_{ij}$ is the sampling options, $\Delta w$ is the sliding window for a given time interval of values and $\Delta w$ equals $n$. |
| 2 | $f\left(\chi^2(P_{in})\right)$ is the statistic value for the chi-squared ($\chi^2$) distribution. | $f\left(\chi^2(P_{in})\right) = \sum\limits_{i=1}^{r} \sum\limits_{j=1}^{c} \frac{(P_{ij} - \overline{P_{ijn}})}{\overline{P_{ijn}}}$ | $P_{ij}$ is the actual frequency in the $i$-th line, $j$-th column; $E_{ij} = \overline{P_{ijn}}$ is the expected frequency in the $i$-th row, $j$-th column; $r$ is the number of rows; and $c$ is the number of columns. |
| 3 | Cumulative function for Poisson distribution [25] $fc(P(P_{in}|\overline{P_{in}}))$ | $fc(P(P_{in}|\overline{P_{in}})) = \sum\limits_{i=1}^{P_{in}} \frac{\overline{P_{in}}^{P_{in}} e^{-\overline{P_{in}}}}{P_{in}!}$ | The cumulative distribution function for Poisson returns the probability that the outcome is less than or equal to $P_{in}$. |

In this study, we use the transformation of each type of distribution using various functions, as described below. In this case, the current value is compared with the average value from the selected interval, which includes the current value. Thus, we can estimate how well the numbers from the given range coincide with the numbers from the expected one [26]. The expected range is taken as the average value obtained for the sample from the values of the sliding window. The estimation range is also chosen based on the absolute values currently appearing in the window. When receiving new values that come from the UAV, the range is shifted by one new value, thus updating the average value and the range of the sliding window. We used the cumulative Poisson distribution function to determine which values fall in the range close to the mean, which are greater, and which are less than it [27]. The statistical value for the $\chi^2$ test allows us to determine the fact of the presence of changes in the parameter under study.

In order to evaluate the values of the distribution, which indicate a potential attack on the UAV, the two conditions are defined as follows:

$$\begin{cases} f\left(\chi^2(P_{in})\right) \to 1 \,(\text{condition 1}) \\ fc(P(P_{in}|\overline{P_{in}})) \to 0.5 \,(\text{condition 1}) \\ f\left(\chi^2(P_{in})\right) \to 0 \,(\text{condition 2}) \\ fc(P(P_{in}|\overline{P_{in}})) \to 0 \,(\text{condition 2}) \\ fc(P(P_{in}|\overline{P_{in}})) \to 1 \,(\text{condition 2}) \end{cases} \tag{6}$$

In Equation (6), Condition 1 corresponds to the case in which there is no attack on the UAV, since there are no critical changes in the parameters. Condition 2 corresponds to the case when critical changes in the parameter are identified, indicating the probability that an attack is being carried out.

Thus, our data normalization technique for UAV classification is as follows.

1. The data collection module receives cyber-physical parameters from the flight controller.
2. The resulting data flow is recorded in the database for further processing.
3. The data normalization module uses Equation (2) from Table 2 to determine the presence of critical changes in the data set.
4. If Condition 1 of Equation (6) is fulfilled for $\chi^2$, then the data set can be classified as normal.

5. If Condition 1 of Equation (6) is not fulfilled for $\chi^2$, then Condition 2 is checked for the cumulative distribution of Poisson according to Equation (3) from Table 2.
6. If Condition 2 for the cumulative distribution of Poisson is fulfilled, then we identify the set of data as abnormal.
7. If Steps 5 and 6 are performed for the latitude, longitude and flight speed parameters, then we can conclude that there is a Spoofing attack in progress [28].
8. If Steps 5 and 6 are performed for the signal level, flight height and flight speed parameters, then we can conclude that there is a Jamming attack in progress [28].
9. The results are written into the database.

## 3. Results

### 3.1. Attack Scenarios

To test our new methodology, two attack scenarios were carried out on two types of UAVs: a home-built UAV based on the PixHawxk flight controller, and a commercial DJI Mavic Air drone.

The first scenario was to spoof the navigation signal to the GPS global navigation system. To carry out the attack, a specialized radio frequency module HackRF One was used. In total, up to 100 test runs were performed. An attacker using specialized equipment, sending a signal of greater power, transmitted the information to the UAV from fake GPS satellites, thereby forcing the UAV to move from a given position. This shift was also accompanied by a change in altitude, the power of the received signal and sometimes crashes of the UAV [29].

The scenarios were run multiple times, and a large amount of data was obtained for analysis. During the experiments for a UAV based on the PixHawxk flight controller, a change in the height of the UAV was observed, as well as a smooth shift of the UAV to the waypoint that was set by the attacker. In a few experiments, when the attack was abruptly interrupted, a fall of the UAV was observed. In some cases, the UAV picked up speed and altitude and abruptly began to move to a given point. In some cases, the UAV was observed to move to a completely opposite point [30]. When conducting this attack on the DJI Mavic Air drone, the following was observed. The number of satellites on which the drone was fixed dropped sharply, usually to four to five satellites, as was revealed by the results of the analysis of the logs, and only satellites of the GLONASS navigation system remained.

The second scenario implemented a denial-of-service attack on the UAV's control channel. If the UAV appears as an access point for connecting using the Wi-Fi protocol, a deauthentication attack on the network can be carried out. This attack is aimed at disconnecting all clients from the access point. The access point resets all connections established with it, and the attack is possible due to the substitution of the MAC address of the access point, the wireless and insecure signal propagation environment and the lack of a mechanism for the authentication and authenticity of this type of message. Thus, a denial of service is provided, i.e., the client loses access to the device.

If another channel or protocol is used to control the UAV, a noise attack using HackRF needs to be used, giving a more powerful signal that "blocks" the legitimate communication channel [31].

As a result of such attacks, legitimate UAV operators lose the ability to control the UAV. This can be assessed by analyzing the logbooks for relevant alerts, as well as by analyzing the signal strength and noise level. During the implementation of the attack, the noise level was significantly exceeded and reached 200 dB, and the signal level range was 0–20%.

When implementing this attack scenario on the DJI Mavic Air, two scenarios were observed. In the first case, the UAV started landing, and even after the attack was over, the operator could not reconnect to the UAV. In the second scenario, the UAV tried to return to the point from which it took off. This happens if the point is originally recorded by the operator for the UAV. In the second case, the operator could recover access to the UAV, but the UAV did not respond to the operator's commands and sought to return to the take-off point.

### 3.2. Analysis of Experimental Data

One of the features of the GPS spoofing attack was a change in the number of satellites that the UAV fixes. In the case of the DJI Mavic Air, the drone did not move to a fake waypoint, and the satellites were jammed. Therefore, we consider this attack GPS jamming. In the case of the Pixhawk flight controller, the attack proceeded as expected. Therefore, let us consider the result of the influence of the attack on the parameters. As can be seen from Figure 2a, the number of satellites fixed (NGS) at the beginning of the attack changed and was unstable, in contrast to normal flight conditions. Figure 2b shows the result of calculating the cumulative function for the Poisson distribution (CDF Poisson). The trusted zone is marked in green, where the values are normal and close to the average value, and they did not change significantly. The red zone is where the observed values are below average, and the CDF tends to be zero. The yellow zone is when values above the mean were observed, and the CDF tends to unity. Thus, it can be seen that the green zone was observed only at the beginning of the flight, when no attack was carried out, and in several places during the flight, when the UAV was fixed on the attacker's satellites. Figure 2c shows the result of calculating the chi-squared ($\chi^2$) function for the NGS. The threshold was set to 0.6, and any values below this were considered abnormal, i.e., the current values are quite far from the mean value of the sample. In the figures where raw data are shown along the vertical axis, the results of the measurements of the cyber-physical parameters are displayed, where the normalized data are the results of calculating functions. The horizontal axis on all graphs displays the number of the time interval.

Figure 3a shows the flight path of the UAV without conducting an attack. As can be seen from the figure, the UAV moves smoothly along a given trajectory, and the coordinates of the sensors coincide with the coordinates of the GPS. Figure 3b shows the flight path during an attack on the UAV. It can be seen from the figure that the UAV deviates from the established trajectory, and the GPS and sensor data do not match.

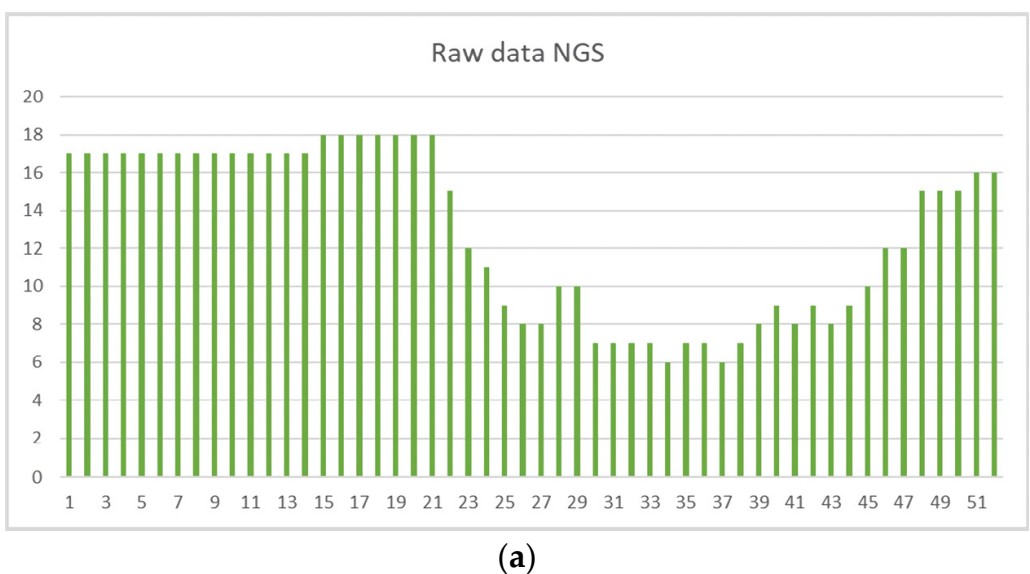

(**a**)

**Figure 2.** *Cont.*

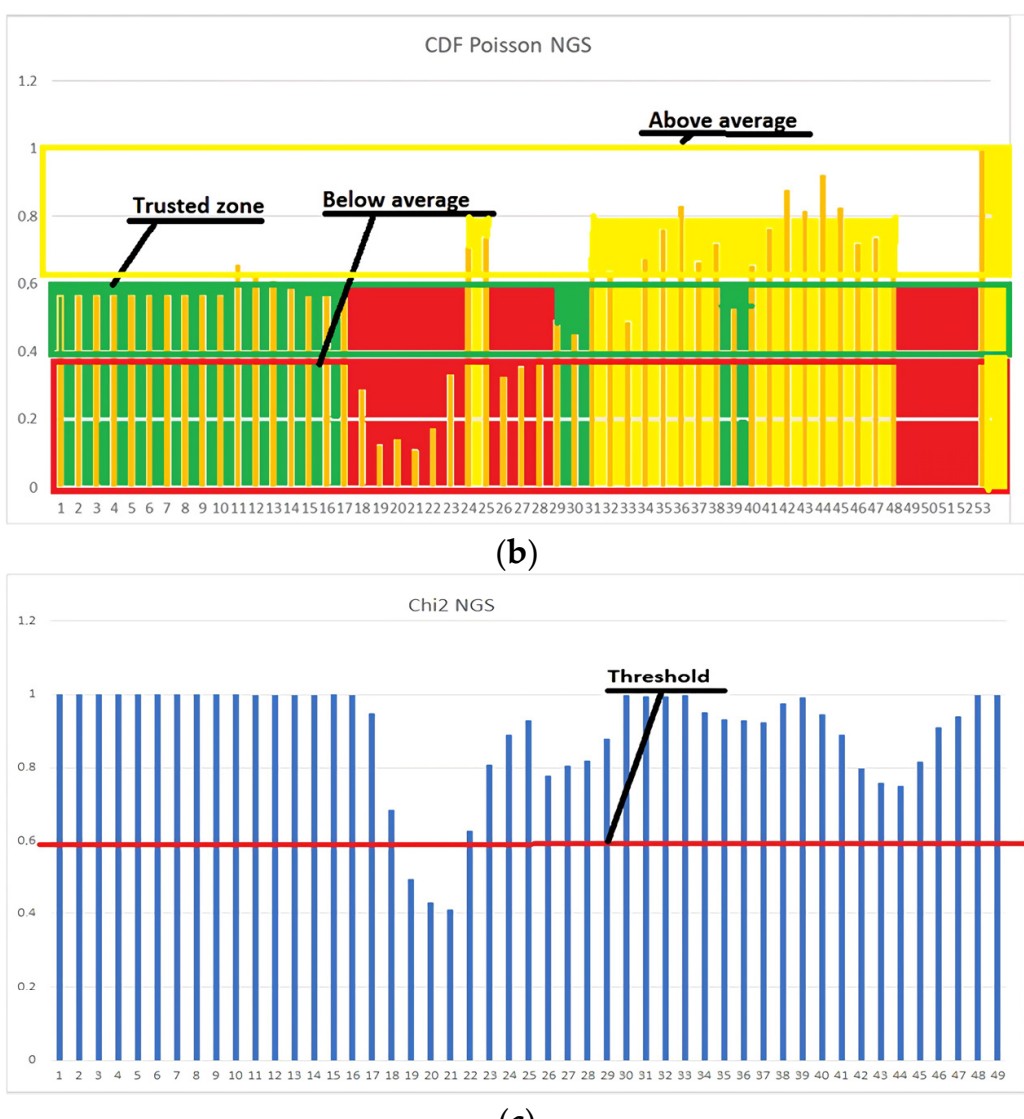

**Figure 2.** The result of the influence of the GPS spoofing attack on the NHS: (**a**) raw data of the number of satellites fixed; (**b**) calculation result of CDF Poisson; and (**c**) calculation result of chi-squared with indications of threshold values.

Let us analyze how the latitude (Lat) and longitude (Long) of the flight changed at the same time. When the UAV moves to the given point smoothly, the parameter also changes gradually, and the results of the calculation of the functions tend to be the values set by Condition 1 in Equation (6). Figure 4 shows the results of the calculations for longitude and latitude. Figure 4a shows that, when the attack was not carried out, the value of the chi-squared ($\chi^2$) test was almost always close to one. This does not mean that the value of the latitude itself did not change. This means that it changed smoothly, and the average value changed gradually. Therefore, the current value was close to it. All data except the value in time range 241 are above the threshold, but even during normal flight, anomalies may occur due to normal changes in flight parameters.

Figure 4b shows that anomalous situations arose for the CDF Poisson latitude, but most of the values tend to the confidence zone. During the attack, it can be clearly seen that there are fewer values that fall into trust zones. If, for the chi-squared ($\chi^2$) test, we are shown only some peaks, when the anomaly is most pronounced, then the CDF constantly changes from a value that is greater than the average to a value that is less than the average. We have similar graphs for longitude. Note that the picture of the graphs is similar for

the number of fixed satellites. Thus, data in different formats can be brought to one. The number of satellites is an integer value in the range of 5–15. The latitude and longitude are floating-point numbers of other orders.

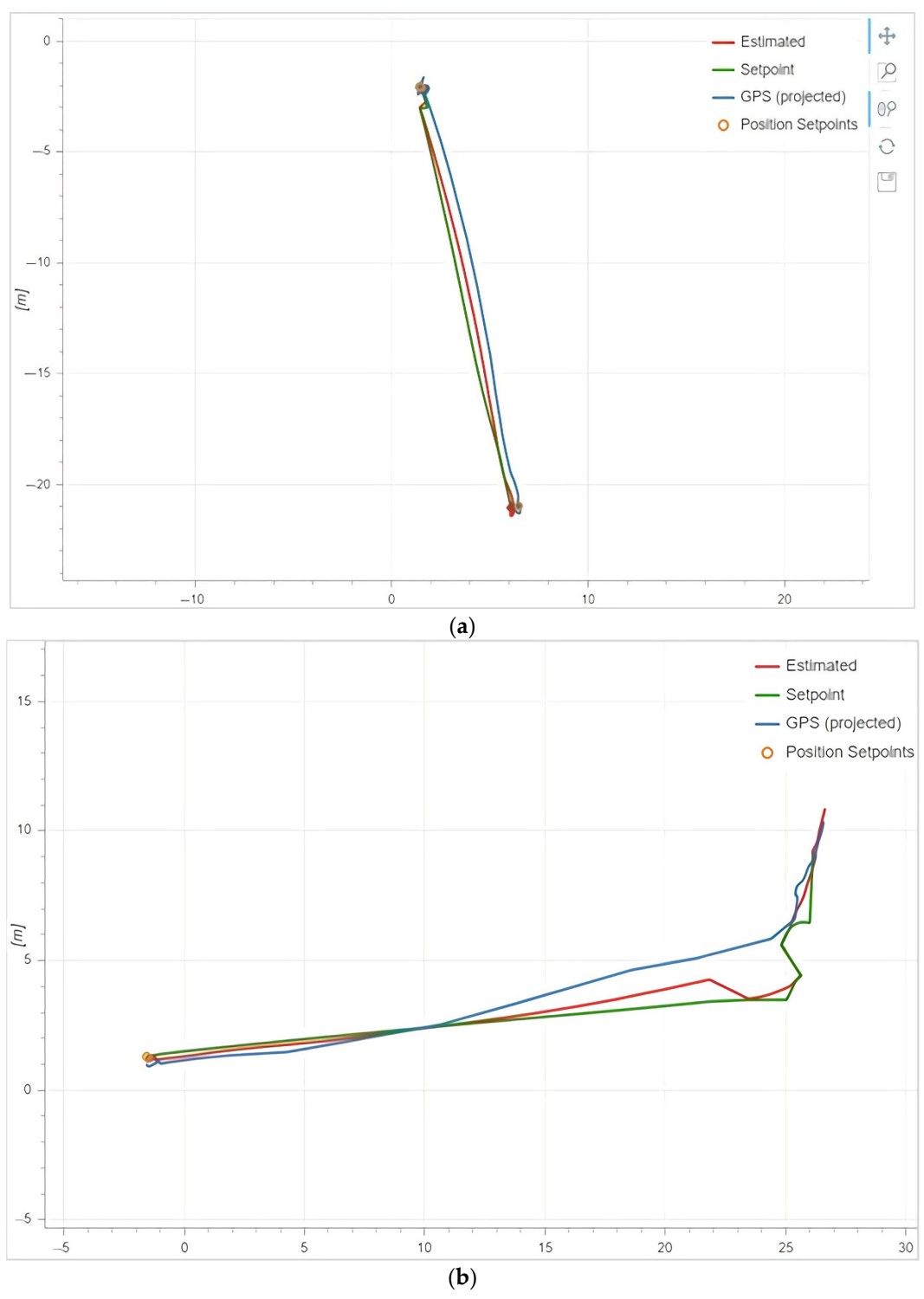

**Figure 3.** The result of the influence of the GPS spoofing attack on the flight coordinates: (**a**) without attack and (**b**) under attack.

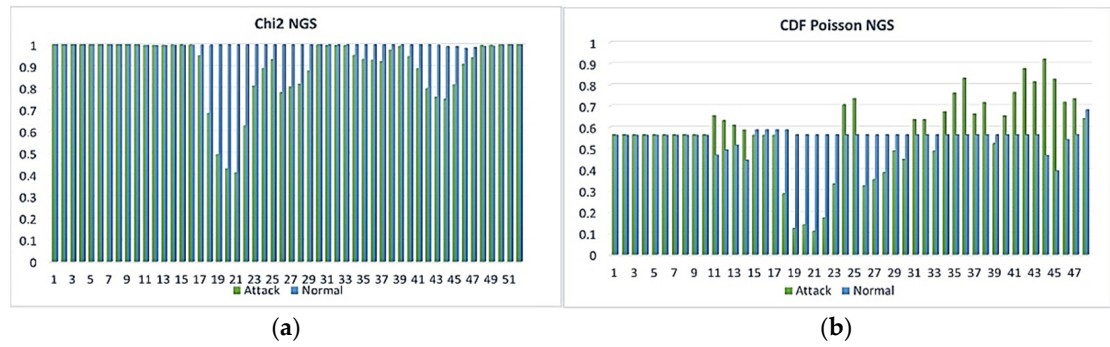

**Figure 4.** The result of the influence of the GPS spoofing attack on the latitude: (**a**) calculation result of chi-squared and (**b**) calculation result of CDF Poisson; and on the longitude; (**c**) calculation result of chi-squared and (**d**) calculation result of CDF Poisson.

Consider a comparison of the number of satellites that the UAV fixes during the attack and without it. Again, the number of satellites can change dramatically and then can remain at the same level, but the transition is recorded. By identifying the jump, we can classify the data as an anomaly. Accordingly, such a set of data can be given for training neural networks to detect an attack. Figure 5 shows that, when there is no attack and the level of the number of satellites fixed is the same, the distribution values are uniform for both cases (a) and (b). The advantage is that the data of a different numerical range lead to one fixed interval.

**Figure 5.** The result of the influence of the GPS spoofing attack on the NGS: (**a**) calculation result of chi-squared and (**b**) calculation result of CDF Poisson.

In addition, as a result of the experimental study, other cyber-physical parameters of the flight were also subject to critical changes under the influence of an attack. This is due to the fact that the attacker can set such parameters by sending a false UAV signal. For instance, during normal flight of a UAV, the UAV's height as measured by a sensor "smoothly" changes a little, but not as much as during an attack. From the raw data of the height (Alt.) in Figure 6c, it can be seen that the UAV's reported altitude decreases and changes as per its actual height under normal conditions. By contrast, the UAV's reported altitude moves abruptly up and down when an attack occurs, even though this may not be its altitude in reality.

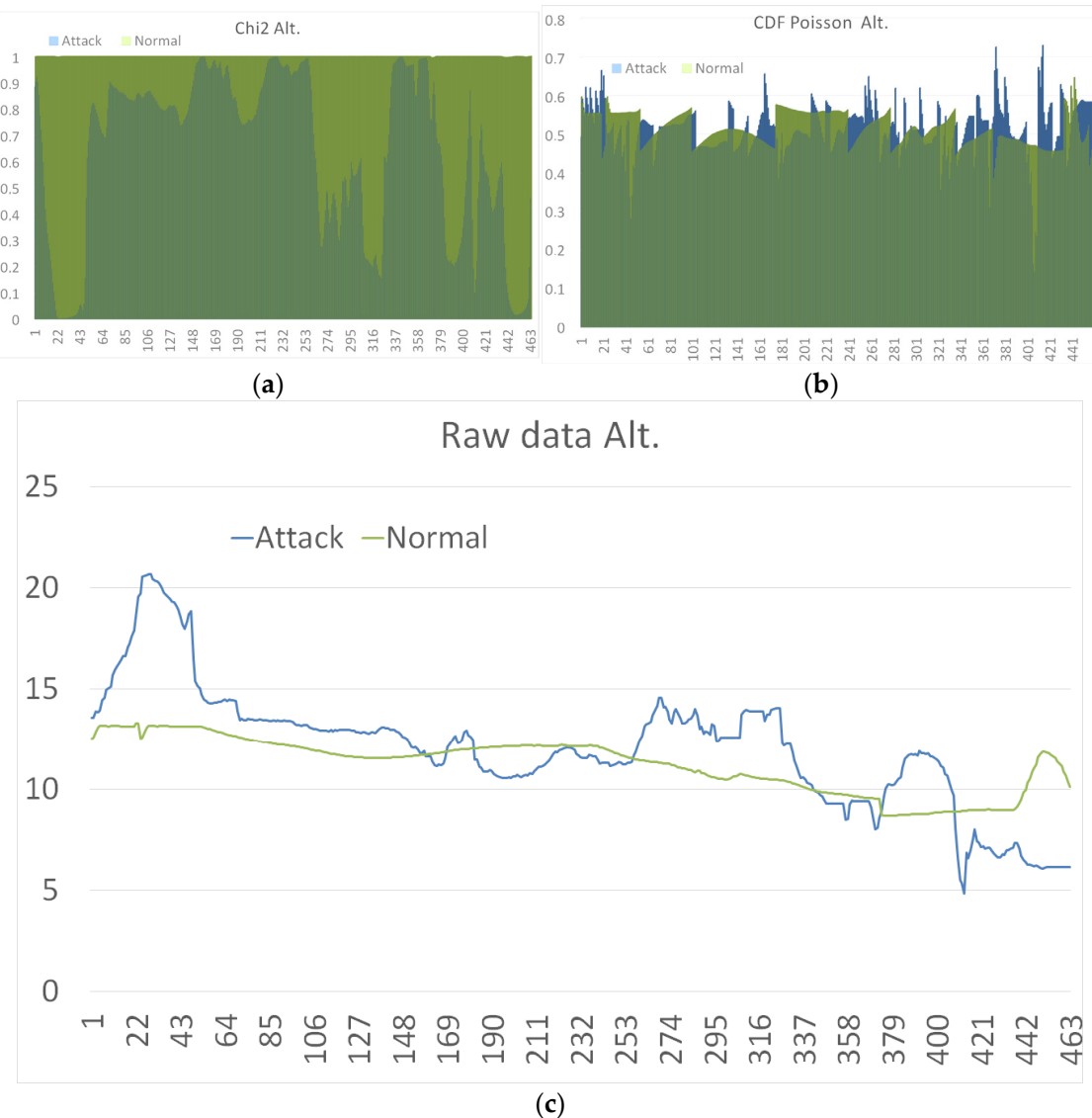

**Figure 6.** The result of the influence of the GPS spoofing attack on the altitude: (**a**) calculation result of chi-squared, (**b**) calculation result of CDF Poisson and (**c**) raw data.

In another experiment, the DJI Mavic Air drone was attacked on the communication channel. At the same time, during the attack, the signal between the UAV and the operator suffered the most disruption. As can be seen from Figure 7, when the signal was red and purple, the signal level was weaker. The signal strength map uses the following colors: green is for a strong signal, orange is for fair signal strength, red is for poor signal strength and purple is for very poor signal strength. Signal strength is calculated based on the presence of a continuous connection to a remote device, and signal loss is detected based

on minor signal interruptions. When there is no attack, the signal is usually green. As can be seen from Figure 7b,c, the signal level is almost the same throughout each flight. It can decrease when the UAV is at a significant distance from the operator, as can be seen especially from Figure 7c, but these values are in the trusted range. When under attack, there is an abrupt decrease and increase in values.

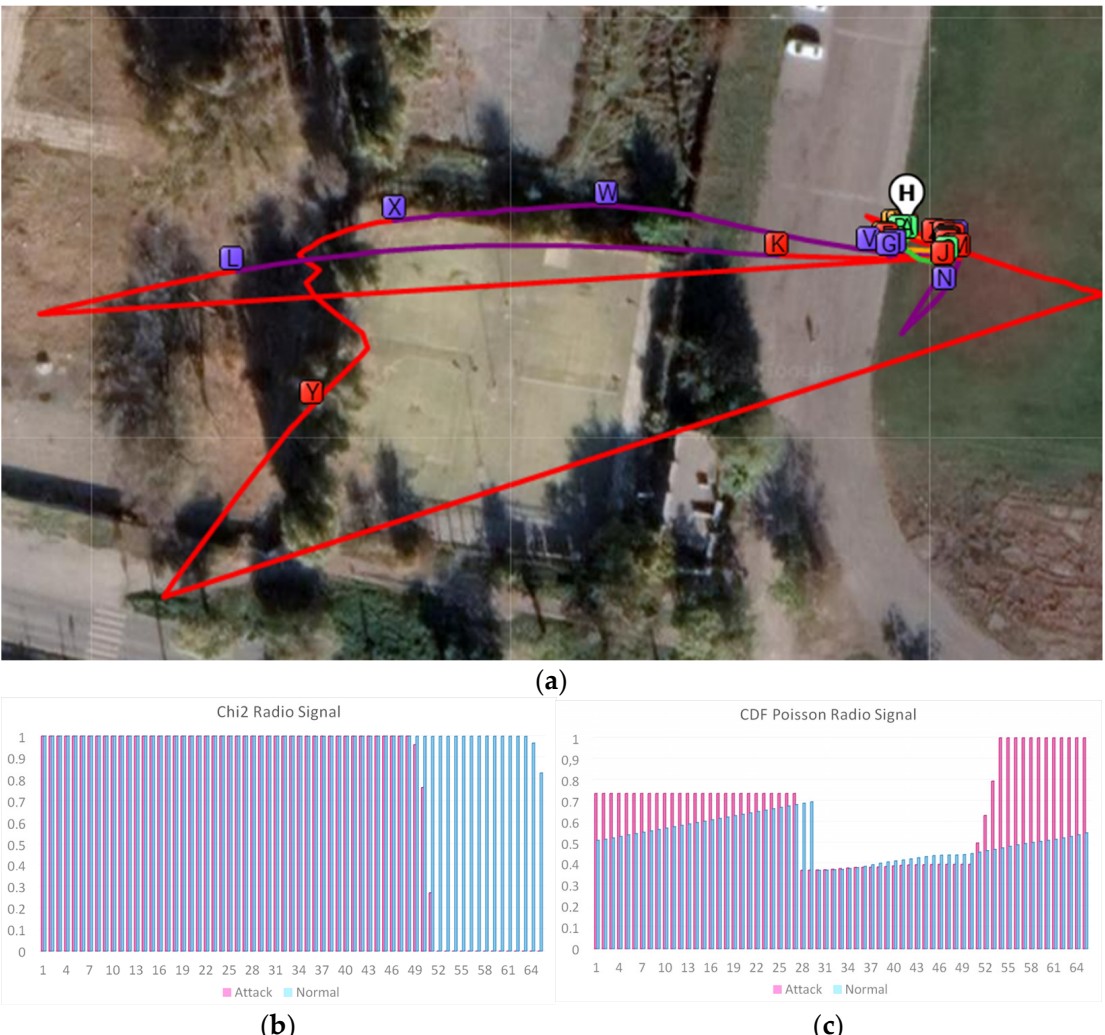

**Figure 7.** The result of the influence of the GPS spoofing attack on the radio signal level: (**a**) raw data, (**b**) calculation result of the chi-squared and (**c**) calculation result of the CDF Poisson.

A similar picture is observed with normalization of the cyber-physical parameter of the speed of flight. UAVs can change their speed occasionally in normal circumstances, as can be seen from the normalized data in Figure 8. This is also noticeable in the graph with raw data. However, the picture is very different when the UAV is under attack, with the reported speed undergoing numerous rapid changes. Some of the accelerations and decelerations shown in the attack data far exceed the UAV's physical capabilities (except, perhaps, the deceleration due to a crash).

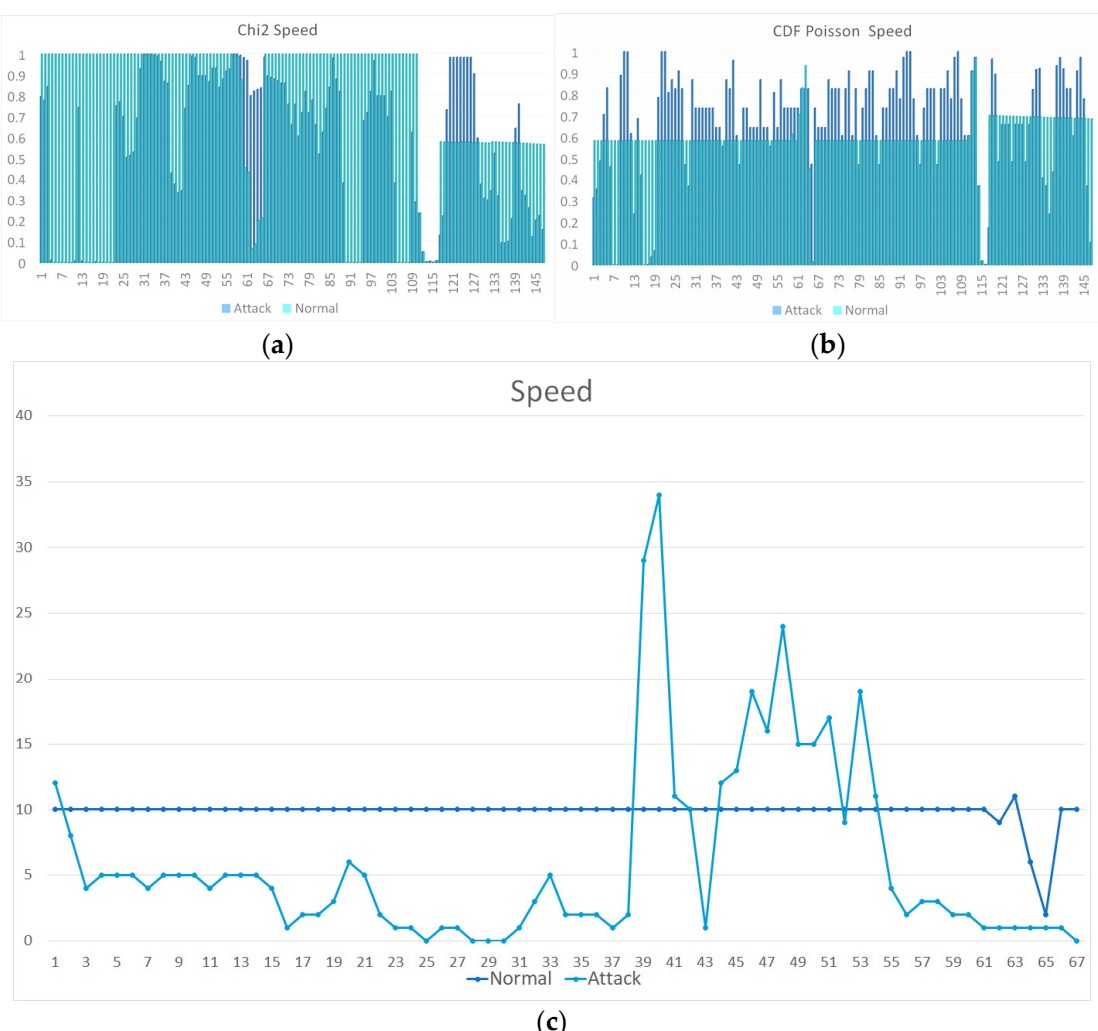

**Figure 8.** The result of the influence of the GPS spoofing attack on the speed: (**a**) calculation result of the chi-squared, (**b**) calculation result of the CDF Poisson and (**c**) raw data.

### 4. Conclusions

This study was aimed not only at the development of a method of normalizing UAV data for further application in an intrusion detection system but also at showing the relationship between specific attacks and cyber-physical parameters, consequences and subsystems. In the articles mentioned in the introduction, usually raw or normalized data were analyzed but only for methods of machine learning, where the probability of detection does not exceed 94%. These normalization methods were applied in our recent studies. It has shown a higher level of detection of up to 99% [29]. Thus, by applying this normalization method, it is possible to increase the level of attack detection, due to sufficiently accurate attack criteria and an unambiguous picture that allows verifying the attack, as was demonstrated in the experimental study.

Thus, in this study, we integrated an ontological approach to assess the impact of various attack categories on the cyber-physical parameters of a UAV. Then, the subsystems of the UAV were related to the cyber-physical parameters. This approach allows simultaneously grouping the signs of an attack based on which subsystem they target. We developed a data normalization technique that can simultaneously allow us to identify the signs of an attack, and, in addition, we defined sets of parameters that signal an attack and described a new database format for intrusion detection for UAVs. Moreover, we performed an experimental study of the impact of attacks on UAV parameters and developed

a software module for collecting data from UAVs, as well as a technique for normalizing and presenting data for detecting attacks on UAVs.

The results obtained can be used to classify attacks, including attack detection systems based on ML methods. In addition, the considered normalization can be used as an independent method for detecting attacks. In this study, we considered normalization in the case of a CPS that includes a UAV. Nevertheless, this method can be used for any kind of CPS. Of course, the Poisson distribution and the chi-squared distribution cannot be applied to all types of data, but we can use the normal distribution in parallel with them. The point is that any CPS should operate in a predictable way. Usually, exceptional events that can be considered as outliers occur rarely. Moreover, when an attack is made, the behavior of the system clearly changes, which leaves a mark on its cyber-physical parameters. In the future, we plan to test the method on data collected from other CPSs.

**Author Contributions:** Conceptualization, E.B. and A.B. (Alexandr Basan); methodology, E.B., A.B. (Alexandr Basan) and E.A.; software, E.B. and A.B. (Alexandr Basan); validation, E.B., A.B. (Alexandr Basan), A.N., C.F., E.A. and A.B (Anatoly Basyuk); formal analysis, E.B. and A.B. (Alexandr Basan); investigation, E.B., A.B. (Alexandr Basan), A.N., E.A. and A.B (Anatoly Basyuk); resources, A.B. (Alexandr Basan); data curation, E.B., A.B. (Alexandr Basan), A.N. and C.F.; writing—original draft preparation, E.B., A.B. (Alexandr Basan), A.N. and C.F.; writing—review and editing, E.B., A.B. (Alexandr Basan), A.N., C.F., E.A. and A.B. (Anatoly Basyuk); visualization, E.B.; supervision, A.B. (Alexandr Basan), E.B. and C.F.; project administration, A.B.; funding acquisition, A.B. (Alexandr Basan). All authors have read and agreed to the published version of the manuscript.

**Funding:** This research was funded by the Russian Science Foundation grant number 22-11-00184, https://rscf.ru/project/22-11-00184/ (accessed on 26 August 2022). The APC was funded by the Russian Science Foundation grant number 22-11-00184.

**Institutional Review Board Statement:** Not applicable.

**Informed Consent Statement:** Not applicable.

**Data Availability Statement:** Data sharing not applicable.

**Conflicts of Interest:** The authors declare no conflict of interest.

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
