# Peer review of "A Data Normalization Technique for Detecting Cyber Attacks on UAVs"

_drones, doi:10.3390/drones6090245_

Round 1

Reviewer 1 Report

This work is undoubtedly a valuable contribution to the issue of aircraft operation and the issue of the safety of unmanned aircraft as well as cyber-physical systems in general.

Please take note of line 422, which is a minor component of a reevaluation. The phrase "self-assembly UAV" doesn't seem acceptable in this situation. It could be better, in my opinion, to swap it out for the term like a "home-built UAV," which would lessen the chance of confusion. 

Why data normalization is given such a strong focus in the study is not quite obvious to me. Personally, I think the benefits of Bayesian statistics should be highlighted more.

Despite not going too far with the concept, the authors provided an excellent formalization of cause (indication) and effect (attack).

The experimental and practical underpinnings of this concept are intriguing and worthwhile.

The arguments put forward are persuasive enough to support the journal publishing of this original work.

Author Response

Dear Reviewer,

Thank you very much for reviewing of our paper, and for your comments and detailed suggestions helped us to improve our paper. We hope that we have understood them well. In accordance with your comments and suggestions as well as suggestions of other reviewers, we have improved our manuscript as much as possible within the allotted time. All the changes/improvements are also highlighted in the Highlighted Changes file.

Reviewer 1:

This work is undoubtedly a valuable contribution to the issue of aircraft operation and the issue of the safety of unmanned aircraft as well as cyber-physical systems in general.

Despite not going too far with the concept, the authors provided an excellent formalization of cause (indication) and effect (attack).

The experimental and practical underpinnings of this concept are intriguing and worthwhile.

The arguments put forward are persuasive enough to support the journal publishing of this original work.

Please take note of line 422, which is a minor component of a reevaluation. The phrase "self-assembly UAV" doesn't seem acceptable in this situation. It could be better, in my opinion, to swap it out for the term like a "home-built UAV," which would lessen the chance of confusion.

Response to Reviewer 1:

We have substituted “self-assembly UAV” by “home-built UAV” to avoid misunderstanding.

Reviewer 1:

Why data normalization is given such a strong focus in the study is not quite obvious to me. Personally, I think the benefits of Bayesian statistics should be highlighted more.

Response to Reviewer 1:

You are right. In the future, we plan to work out the statistics of Bayes and expand the study in this direction. In this study, the main goal was to improve the quality of data sets for teaching neural networks. Including the main task was the definition of signs of attack, which were successfully managed to identify through normalization.

We have added the following paragraph after Equation (5):

“From Bayes’ theorem and probability, in the future, we can proceed to the construction of a Bayesian classifier to determine the types of attacks by changing parameters. The Bayesian classifier belongs to the ML category of algorithms. The bottom line is this: the system, which is faced with the task of determining whether the next change in the parameter is anomalous, has been trained in advance by some amount of data, in which it is already determined where the anomaly is, and where the normal behavior is. It has already become clear that this is training with a teacher, where we act as a teacher. The Bayesian classifier represents the data set that we receive from the UAV (in our case, these are changes in cyber-physical parameters) in the form of a set of metrics that supposedly do not depend on each other (this is where the very naivety comes from). It is necessary to calculate the score for each class (normal behavior/attack spoofing attack silencing) and choose the one that turned out to be the maximum.”.

Also, some other improvements have been made in the article in accordance with the suggestions of other reviewers. They are also highlighted in the Highlighted Changes file.

Reviewer 2 Report

The authors developed a data normalization technique that additionally allows us to identify the signs of a cyber attack. In addition, the authors performed an experimental study of the impact of attacks on UAV parameters and developed a software module for collecting data from UAVs, as well as a technique for normalizing and presenting data for detecting attacks on UAVs. I would like to thank the editor to give me the opportunity to review this interesting work. The impression of the paper is interesting. I have a few minor comments.

+The work proposed and methodology are not well explained.

+In the introduction section, the contributions are not well explained.

+the abstract needs to be re-write to highlight the novelty of the study well with a list of some of the results.

+Figures resolution needs to improve

+English has to be improved to overcome some mistakes.

Author Response

Dear Reviewer,

Thank you very much for reviewing of our paper, and for your comments and detailed suggestions helped us to improve our paper. We hope that we have understood them well. In accordance with your comments and suggestions as well as suggestions of other reviewers, we have improved our manuscript as much as possible within the allotted time. All the changes/improvements are also highlighted in the Highlighted Changes file.

Reviewer 2:

The authors developed a data normalization technique that additionally allows us to identify the signs of a cyber attack. In addition, the authors performed an experimental study of the impact of attacks on UAV parameters and developed a software module for collecting data from UAVs, as well as a technique for normalizing and presenting data for detecting attacks on UAVs. I would like to thank the editor to give me the opportunity to review this interesting work. The impression of the paper is interesting. I have a few minor comments.

+The work proposed and methodology are not well explained.

Response to Reviewer 2:

We have added the following paragraph at the end of subsection 2.3:

“Thus, our data normalization technique for UAV classification is as follows.

  1. The data collection module receives cyber-physical parameters from the flight controller.
  2. The resulting data flow is recorded in the database for further processing.
  3. The data normalization module uses Equation (2) from Table 2 to determine the presence of critical changes in the data set.
  4. If Condition 1 of Equation (6) is fulfilled for χ2, then the data set can be classi-fied as normal.
  5. If Condition 1 of Equation (6) is not fulfilled for χ2, then Condition 2 is checked for the cumulative distribution of Poisson according to Equation (3) from Table 2.
  6. If Condition 2 for the cumulative distribution of Poisson is fulfilled, then we identify the set of data as abnormal.
  7. If Steps 5 and 6 are performed for parameters latitude, longitude, and flight speed, then we can conclude there is a Spoofing attack in progress [29].
  8. If Steps 5 and 6 are performed for parameters signal level, flight height and flight speed, then we can conclude there is a Jamming attack in progress [29].
  9. Write the results into the database.”.

Also, we have added the following paragraph at the end of section Introduction:

“The contributions of this article are as follows. Firstly, an analysis of attacks was carried out that are most characteristic of UAVs, as a cyber-physical system. It is determined what cyber-physical parameters are affected by various attack classes. Secondly, formalization and modeling of UAVs, as a CPS, through the analysis of the interconnection of UAV subsystems to cyber-physical parameters and their changes was carried out. Thirdly, a technique for normalizing cyber-physical parameters is presented. Bringing various types of data to one format makes it easier to meet the conditions for analysis of their changes and evaluating the quality of the parameter (whether the parameter changes normally, or sharp abnormal changes are observed). The formal description of the UAVs presented in the article as a CPS can become the basis for creating a digital twin of UAVs and researching it in terms of cyber security. The presented normalization technique will allow processing raw data, as well as classifying data sets for the use of ML methods in the future.”.

Reviewer 2:

+In the introduction section, the contributions are not well explained.

Response to Reviewer 2:

We have added the following paragraph at the end of section Introduction:

“The contributions of this article are as follows. Firstly, an analysis of attacks was carried out that are most characteristic of UAVs, as a cyber-physical system. It is determined what cyber-physical parameters are affected by various attack classes. Secondly, formalization and modeling of UAVs, as a CPS, through the analysis of the interconnection of UAV subsystems to cyber-physical parameters and their changes was carried out. Thirdly, a technique for normalizing cyber-physical parameters is presented. Bringing various types of data to one format makes it easier to meet the conditions for analysis of their changes and evaluating the quality of the parameter (whether the parameter changes normally, or sharp abnormal changes are observed). The formal description of the UAVs presented in the article as a CPS can become the basis for creating a digital twin of UAVs and researching it in terms of cyber security. The presented normalization technique will allow processing raw data, as well as classifying data sets for the use of ML methods in the future.”.

Reviewer 2:

+the abstract needs to be re-write to highlight the novelty of the study well with a list of some of the results.

Response to Reviewer 2:

We have improved the abstract by adding the following text at its end.

“Data analysis and evaluation of the quality of a parameter (whether the parameter changes normally, or abrupt anomalous changes are observed) is facilitated by converting different types of data to the same format. The resulting formalized CPS model allows us to identify the nature of an attack and its potential impact on UAV subsystems. In the future, such a model could be the basis of a CPS digital twin in terms of security. The presented normalization technique supports processing raw data, as well as classifying data sets for their use in machine learning (ML) analyses in the future. The data normalization technique can also help to immediately determine the presence and signs of an attack, which allows classifying raw data automatically by dividing it into different categories. Such a technique could form the basis of an intrusion detection system for CPSs. Thus the obtained results can be used to classify attacks, including attack detection systems based on machine learning methods, and the data normalization technique can be used as an independent method for detecting attacks.”

Reviewer 2:

+Figures resolution needs to improve

Response to Reviewer 2:

The resolution has been improved in a MS Word file. Probably, it decreases due to file conversation to a pdf file.

Reviewer 2:

+English has to be improved to overcome some mistakes.

Response to Reviewer 2:

English has been checked and corrected again by a native English-speaking co-author.

Also, some other improvements have been made in the article in accordance with the suggestions of other reviewers. They are also highlighted in the Highlighted Changes file.
